# Chronic and Acute Toxicities of Aflatoxins: Mechanisms of Action

**DOI:** 10.3390/ijerph17020423

**Published:** 2020-01-08

**Authors:** Noreddine Benkerroum

**Affiliations:** Department of Food Science and Agricultural Chemistry MacDonald Campus, McGill University, 21111 Lakeshore, Ste Anne de Bellevue, QC H9X 3V9, Canada; n.benkerroum@gmail.com; Tel.: +1-514-652-4945

**Keywords:** aflatoxins, tumorigenicity, carcinogenicity, acute toxicity, immunogenicity, genotoxicity

## Abstract

There are presently more than 18 known aflatoxins most of which have been insufficiently studied for their incidence, health-risk, and mechanisms of toxicity to allow effective intervention and control means that would significantly and sustainably reduce their incidence and adverse effects on health and economy. Among these, aflatoxin B1 (AFB1) has been by far the most studied; yet, many aspects of the range and mechanisms of the diseases it causes remain to be elucidated. Its mutagenicity, tumorigenicity, and carcinogenicity—which are the best known—still suffer from limitations regarding the relative contribution of the oxidative stress and the reactive epoxide derivative (Aflatoxin-exo 8,9-epoxide) in the induction of the diseases, as well as its metabolic and synthesis pathways. Additionally, despite the well-established additive effects for carcinogenicity between AFB1 and other risk factors, e.g., hepatitis viruses B and C, and the hepatotoxic algal microcystins, the mechanisms of this synergy remain unclear. This study reviews the most recent advances in the field of the mechanisms of toxicity of aflatoxins and the adverse health effects that they cause in humans and animals.

## 1. Introduction

Aflatoxins are the mycotoxins of the greatest concern to food safety due to their wide distribution in foods and feeds and their high toxicities. Since their discovery, aflatoxins have been associated with liver cancer, with peanut, maize and their derivatives being the main vehicles. Geographically, tropical and subtropical regions are the most affected by aflatoxins as food and feed contaminants and as chemical hazards that contribute greatly to the high incidence of a number of devastating chronic diseases and aflatoxicosis outbreaks.

Due to the public health concerns that these toxicants raise and their association with genotoxic effects, intensive studies have been carried out since their discovery to elucidate the mechanisms of their carcinogenicity and other toxicities. The elucidation of these toxicological aspects is a prerequisite to the design of curative or preventive means, and to adequately regulate their occurrence in foods and feeds.

Although aflatoxins have primarily been associated with cancer diseases, it is now well established that they cause various other acute and chronic diseases, most of which are severe. The carcinogenicity of aflatoxins has long been associated with the liver, where they are first metabolised to release reactive intermediate metabolites. However, subsequent epidemiological and animal studies demonstrated their carcinogenicity to remote organs from the liver, including the kidney, the pancreas, the bladder, bone, viscera, central nervous system, etc. [1]. In the case of aflatoxin B1 (AFB1), the best studied aflatoxin, AFB1-exo 8,9-epoxide, which is formed as the first step of AFB1 metabolism by microsomal cytochrome enzyme (CYP450) has been considered as the ultimate responsible for genotoxicity. Currently, there is increased evidence that the oxidative stress caused by AFB1 plays an equal or even higher role in the genotoxicity of the aflatoxin. Immunotoxicity of aflatoxins is probably the second most documented toxicological effect and its mechanisms of action are being increasingly elucidated. Aflatoxins are prominently immunosuppressive but immunostimulatory actions have been reported. In addition to these leading toxicological effects, other aflatoxin-induced acute and chronic health issues, such as malnutrition diseases, retarded physical and mental maturity, reproduction, nervous system diseases, etc. have been demonstrated in humans or animals, but their mechanisms of actions require further clarifications [2].

This work aims to present an up-to-date overview of the adverse acute and chronic health effects caused by aflatoxins and their mechanisms of action with an emphasis on the most recent discoveries in the field.

## 2. Chronic Diseases Caused by Aflatoxins

Repeated exposure to low doses of aflatoxins over a lifetime causes chronic diseases, the most frequent and severe of which is cancer. Although dietary intake of aflatoxins has been classically associated with primary liver cancer, i.e., HCC and bile duct hyperplasia [3], other organs, such as the kidney, the pancreas, the bladder, bone, viscera, etc., have also been reported to develop cancer upon exposure to these mycotoxins [1]. In addition, aflatoxins were reported to cause lung [4] and skin [5] occupational cancers via inhalation and direct contact, respectively. In fact, chronic exposure to aflatoxins causes a range of other severe diseases, including immunosuppression, teratogenicity, mutagenicity, cytotoxicity, and estrogenic effects in mammalians [6]. Moreover, aflatoxins are believed to be involved in nutritional disorders, such as kwashiorkor and growth faltering, probably by interfering with the absorption of micronutrients (e.g., zinc, iron, and vitamins), protein synthesis, and metabolic enzyme activities [2,7]. In domestic animals, feeds contaminated with sub-lethal doses of aflatoxins induce impaired productivity and reproduction, increased susceptibility to diseases, and reduced quality of the foods they produce [8]. Despite the insidious character of chronic aflatoxin-induced diseases, their impact on public health globally is more severe and more costly than acute aflatoxicosis. Although, the aflatoxicosis outbreaks induce hundreds of deaths at once in an intermittent manner, they can be prevented or interrupted upon analysis of suspect crops/foods, e.g., evident mould growth, and their disposal if aflatoxin levels exceed the regulatory standards.

Liver cancer is one of the most common and deadly type of cancer diseases whose occurrence has been strongly correlated with dietary exposure to aflatoxins, which is enhanced in the presence of other risk factors [2]. Notably, chronic infections with hepatitis virus B (HB) were shown to cause an increase in the potency of AFB1 by up to 60 times [9]. According to the most recent statistics given by the global cancer observatory of the IARC (http://gco.iarc.fr, accessed on 1 September 2019), 841,080 new cases of liver cancer causing 781,631 deaths were recorded globally in 2018. This corresponds to an age-standardized incidence rate of 9.3 per 100,000 and mortality rate of 93% ranking as the fifth cancer type and the first cause of cancer-induced mortality. Africa and Asia continue to be the leading continents in terms of new cases recorded each year, with 64,779 (7.7%) and 609,596 (72%) cases respectively, together representing about 80% of the total cases in the world. Aflatoxin B1 alone was estimated to cause 25,200 to 155,000 cases each year [10,11], 40% of which occur in sub-Saharan Africa only [2] where aflatoxin-induced liver cancer accounts for one-third of all liver cancer cases recorded in the whole African continent [12]. At the country level, China has the highest incidence of liver cancer in the world, with the vast majority being recorded in the Southern part of the country where the two main synergistic causative agents, exposure to dietary aflatoxins and HB chronic infections, are endemic and highest [3].

## 3. Mechanisms of Toxicity at Glance:

Aflatoxins exert various toxicological effects with different mechanisms, most of which are not yet fully elucidated. Intensive research has been carried out to investigate the mechanisms of the toxicity of aflatoxins to provide a scientific basis for the design of preventive and control means. Advanced knowledge in the field can also serve as a scientific tool for the provision of regulatory purposes by food safety authorities. The mutagenic effects of AFB1 have been the focus of most studies since the discovery of this aflatoxin and were ascribed mainly to the intermediate metabolite AFB1-exo-8,9 epoxide (AFBO) [13]. As a highly unstable molecule, AFBO reacts with cellular macromolecules, including nucleic acids, proteins, and phospholipids, to induce various genetic, metabolic, signalling, and cell structure disruptions [14,15,16]. However, increased evidence is being built up demonstrating equally dramatic or higher effects of AFB1 on cell function and integrity through the induction of oxidative stress (OS) [17,18,19]. Figure 1 summarizes the different toxicity mechanisms of AFB1 involving AFBO and OS to cause genotoxicity, immunotoxicity and acute intoxication by acting on genomic DNA, other functional macromolecules and immunocompetent cells.

## 4. Genotoxicity and Cancer Diseases

### 4.1. AFBO-Mediated Genotoxicity

AFBO has long been considered as the ultimate metabolite responsible for the genotoxic effects of AFB1 as well as other aflatoxins bearing a double bond between carbons C^8^ and C^9^ done in the furan ring [6,21]. The mechanisms of toxicity mediated by this AFB1 reactive metabolite are the best understood and have been extensively reviewed [17,22,23,24]. Upon ingestion, AFB1 is absorbed in the duodenum and reaches the liver where it is bioactivated by the action of various microsomal cytochrome enzymes (CYP450). These are monooxygenases that catalyse the oxidation of the C^8^ = C^9^ double bond in the furan ring yielding AFB1-exo and -endo 8,9 epoxide stereoisomers, with the former isomer being >1000 times more reactive/toxic than the latter [25]. Different CYP450 isozymes are responsible for the bioactivation of AFB1 depending on the host, the organ, and the sub-cellular component. In humans, among 57 CYP450 identified isoenzymes, the microsomal CYP1A2, 3A4, 3A5, 3A7, 2A3, and 2B7, the hepatocytic 3A3, and the lung CYP2A13 are the principal isozymes responsible for AFB1 bioactivation in the respective organs [26,27]. In the liver, the bioactivation is essentially catalysed by CYP1A2 or 3A4, with predominating action under low or high exposure conditions, respectively; CYP1A2 predominates under the actual food contamination levels and is also responsible for the transformation of AFB1 into the less toxic endo-epoxide isomer [28,29]. In animals and insects, various CYP450 isozymes, including CYP1A1, 1A, 1A2, 2A5, 2A6, 3A, 3A4, 3A13, and 321A1, were reported to catalyse the bioactivation step, depending on the species and the organ where they are produced. The specific roles of CYP450 isozymes in AFB1 metabolism and their distribution in different hosts and organs were reviewed elsewhere [30].

Once released, AFBO intercalates the DNA and binds covalently, upon alkylation reaction, to the N^7^ atom of guanine residue forming a stereospecific aflatoxin-DNA adduct, *trans*-8,9-dihydro-8-(N^7^-guanyl)-9-hydroxy-AFB1 (AFB1-N^7^-gua) [23], most frequently (60–80%) at the third guanine residue of the codon 249 (^5′^-AG*G-^3′^) on *p53*/*PT53* tumour suppressor gene [31,32]. Due to its positive charge, AFB1-N^7^-gua adduct is highly unstable and releases itself leaving an apurinic DNA molecule (AP). However, the imidazole ring may be opened under slightly alkaline conditions to form two stable isomers *cis*- and *trans*-AFB1-formamidopyrimidine (AFB1-FAPy) adducts, also called minor and major AFB1-FAPy adducts, respectively (Figure 1 and Figure 2). These three AFBO-induced DNA lesions (AP, AFB1-N^7^-gua, and AFB1-FAPy) have been known as the main precursors of AFB1 genotoxic and carcinogenic effects (Figure 1). Among them, AFB1-FAPy was reported to be the most mutagenic due to its persistent DNA damage [33,34], which was ascribed to the less helix-distorting lesions it induces compared with those caused by AFB1-N^7^-gua, thereby hindering DNA repair [17,22]. AFB1-FAPy lesions are essentially repaired by the nucleotide excision repair (NER) mechanism, which is contingent to the extent of DNA helix distortion for the recognition of damaged sites; the more distorted the site, the easier it is to be recognized by the repair proteins [17,35,36].

### 4.2. Genetic Polymorphism and Increased Mutagenicity of Aflatoxins

The higher mutagenicity of AFB1-FAPy lesions may not be explained solely by its refractory behavior to NER repair, as they can also be repaired by the less helix-distortion sensitive mechanism of base excision repair (BER) [37]. BER involves a site-specific recognition by glycosylases of the damaged bases to be excised and then replaced by the correct base [38]. However, it is now well established that exposure to aflatoxins induces various epigenetic changes in repair genes that impedes BER. For example, hypermethylation of the promoter of *NEIL1* (Nei Like 1) gene coding for a DNA glycosylase (NEIL1), which plays a key role in BER, was recently shown to reduce the excision efficiency in AFB1-FAPy adducts by transcriptional repression of the gene [39]. The repair of AFB1-FAPy lesions may be further restricted in humans due to the widespread polymorphic variants producing catalytically inactive NEIL1 enzyme [40]. Polymorphism in other human DNA repair genes, such as *XPC*, *XPD*, *XRCC1*, *XRCC3*, *XRCC4*, *XPD* and *XRCC7*, has been reported to be an additional factor that increases the risk of aflatoxin-induced HCC in high exposure environments [38,41,42,43,44]. This risk is exacerbated with simultaneous polymorphism of repair genes and phase II-enzyme detoxifying genes, as was demonstrated for the combined polymorphisms of *XRCC1* (involved in BER repair) with *GSTM1* and *HYL1*2* (coding for GST and microsomal epoxide hydrolase, respectively) [43,44]. Conversely, the effect of AFB1-detoxifying gene polymorphism alone on the increase in HCC risk remains controversial [43,44,45,46,47,48,49]. It should be pointed out, however, that most of the reports on the interaction of polymorphisms with AFB1 exposure to increase HCC risk were case–control studies conducted on highly exposed populations in Guangxi (China) and The Gambia [38,49]. The rationality of these studies suffered biased uncertainties due to limited access to HCC-case patients and the possible interference with other factors, such as smoking, drinking, and impaired liver functionality of HCC cases yielding imprecise biomarker estimates. Nevertheless, there is a consensus on the likely interaction between exposure to AFB1 and polymorphism of the repair genes to increase HCC risk, especially in high-risk environments, e.g., high exposure and chronic hepatitis virus infections. Moreover, the higher resistance to DNA repair of AFB1-FAPy adduct was attributed to its ability to stabilize the double helix—owing to the way of its insertion between the helices [35]. Once intercalated between the DNA helices at the G:C site, FAPy stacks with neighboring base pairs to stabilize the double helix, which is enhanced in the presence of the formamido group that probably establishes intra-strand sequence-specific hydrogen bonding within the helix [50]. Nevertheless, irrespective of the lack of a clear mechanistic explanation, various observations and mutational studies on the stability of lesions and repair efficiencies have established the implication of the AFB1-FAPy adduct in the vast majority of AFB1-induced mutations and, therefore, its higher genotoxicity compared with the other AFBO-induced lesions [17,22,23,51].

### 4.3. Cell Cycle Progress as Affected by Aflatoxin-Induced p53 Gene Mutation

Failure to repair the DNA damaged by any of the above-mentioned lesions leads to transversion mutations, predominantly G→T (80%), of the third base ^5′^G of the codon 249 on *p53* gene; in few instances, the second base of the codon or G→A transversions have been reported [22,32]. As a result, the mutant expresses a non-functional protein where the serine residue at the position 249 is substituted for arginine. The resulting altered protein, pR249S, cannot bind to DNA molecules and hence loses its transactivation capacity towards a multitude of *p53*-dependent gene promoters responsible for various vital cellular functions, including cell cycle arrest, senescence, and apoptosis, eventually leading to tumorigenicity [53,54]. Two of the most known *p53*-dependent regulatory genes involved in cell cycle progression and/or apoptosis signalling pathways are *CDKN1A* (cyclin-dependent kinase inhibitor 1A) and *PUMA* (*p-53* upregulated modulatory apoptosis). In normal functioning conditions, exposure to genotoxic insults upregulates the latter genes by the transcriptional factor p53, to express the effector proteins CDKN1A/p21 and PUMA, respectively (Figure 3).

p21, also known as p21^WAF1/Cip1^, regulates negatively the progress of cell division mainly through the inhibition of cyclin-dependent kinases (CDKs) or proliferating cell nuclear antigen (PCNA) as illustrated in Figure 3. Various mechanisms have been proposed for p53-driven cell cycle arrest emphasizing the role of CDKN1A gene product (p21 protein) which antagonizes with CDKs responsible for the initiation of a cascade of events leading to the repression of many genes involved in the progress of the cell cycle at different checkpoints [54,58,59]. A more recent mechanism suggests an indirect repression of cell cycle progression via the p21-DREAM-E2F/CHR (p53-DREAM) pathway, wherein a transcriptional repressor, DREAM (dimerization partner, RB-like, E2F and multivulval class B), binds to E2F and CHR (cell cycle gene homology region) promoter sites and downregulates the transcription of more than 250 genes controlling the progression of the whole cell cycle at different stages from G0 to cytokinesis [55,56]. DREAM is a multi-subunit complex composed of a core multivulval class B (MuvB) complex associated with E2F4 or E2F5, dimerization partner (DP), and proteins p107 or p130 (also called RB-like proteins 1 and 2, respectively) (Figure 3A). However, for p107 and p130 to bind and activate the other subunits of the DREAM complex, they should be in their hypo-phosphorylated states, which requires active p21 to inhibit cyclin-CDK complexes, e.g., cyclinE-CDK2 and cyclin D-CDK4/6, responsible for their hyper-phosphorylation [55]. Sequestration of PCNA by p21 can also evoke cell cycle arrest at a given stage of the cycle by blocking DNA replication and repair requiring PCNA as a co-factor for the activity of DNA polymerase [58] (Figure 3B). Although these studies have been carried out on different cell types and organs, and with different DNA-damaging or -nondamaging stimuli, the results can apply to AFB1-induced DNA damage. An intraperitoneal administration of AFB1 to mice at a daily dose of 20 µg/kg bw for up to 21 days induced overexpression of p21 with concomitant downregulation of cyclin D1 and CDK4, which inhibited the formation of cyclin-CDK complexes, ultimately leading to cell cycle arrest and apoptosis [62]. However, the study demonstrated that although both cell cycle arrest and apoptosis were *p53*-dependent, the upregulation of p21 expression was not involved in apoptosis. In fact, it is well established that p21 plays an antagonistic dual role, not yet well understood, as it can either restrict or promote apoptosis depending on many factors, such as the extent of DNA damage, the type of stimulus, the tissue, the type of cell line, the subcellular localisation, chemotherapy treatment (if any), etc. [59]. In the cytoplasm, p21 primarily exerts an anti-apoptotic, i.e., tumour promoting, action whereby it inhibits key enzymes responsible for the induction of apoptosis or the transcription factors responsible for the transactivation of genes coding for pro-apoptotic proteins (Figure 3C). The anti-apoptotic effect while the cell cycle is arrested needs further clarifications for the circumstances of its occurrence and regulatory mechanisms that trigger the switch from pro-apoptotic to anti-apoptotic action and vice-versa. Presently, the prevailing explanation considers that, in conjunction with the cell cycle arrest, p21 inhibits apoptosis to ensure cell survival during the pause of cell cycle arrest in order to provide an opportunity for DNA repair before proceeding with the normal growth cycle. However, in case of a severe damage and impaired DNA repair, the role of p21 located in the nucleus is switched to be pro-apoptotic and activates the caspase cascade driving cell death [57,58,59].

Cells exposed to various genotoxic and nongenotoxic stimuli may undergo *p53* dependent or independent apoptosis; however, genotoxic stimuli, e.g., exposure to AFB1, causing severe DNA damage trigger essentially *p53*-dependent apoptosis, involving two main regulatory proteins of the family of the Bcl-2 homology domain 3 (BH3)-only, PUMA and NOXA, with PUMA being involved in virtually all *p53*-dependent apoptotic activities [63]. PUMA is transcriptionally upregulated by p53 protein to antagonize with pro-survival proteins of the Bcl-2 family that inhibit constitutively the pro-apoptotic pore-forming BAX (Bcl-2-associated X protein) and/or BAK (Bcl-2 antagonist/killer) proteins. The inhibition of the pro-survival proteins activates BAX/BAK which oligomerize to form pores in the outer membrane of mitochondria allowing leakage of pro-apoptogenic proteins that initiate a cascade of events ending with the activation of caspases directly involved in cell death (Figure 3D).

Under DNA damaging conditions in *p53* mutants, *CDKN1A* and *PUMA* genes remain repressed due to the lack of functional p53 transcriptional factor. Consequently, CDKs, PCNA, and pro-survival Bcl-2 proteins are relieved, leading to an uninterrupted cell cycle with poor repair machinery and an overexpression of *CDK* genes, among multiple other physiological dysfunctions (Figure 3).While the restriction of cell cycle arrest increases the likelihood for unfaithful DNA replication and the accumulation of mutations among other metabolic and signaling dysfunctions, the overexpression of CDKs was shown to play crucial role in tumor development and promotion [64,65]. Restriction of apoptosis, a major physiological function for the elimination of senescent, damaged, or stressed cells, due to the repression of PUMA and cytoplasmic p21 in *p53* mutants, exacerbates the risk for cancer diseases. It is also well established that any disruption in the expression and the signaling pathways involving pro-apoptotic or pro-survival proteins of the Bcl-2 family members not only promotes cancer but also increases its resistance to chemotherapy [66]. In fact, in addition to the restriction of cell cycle arrest and apoptosis, *p53* mutations de-regulate the genetic expression of a plethora of genes controlling various other cellular functions and metabolic pathways, such as the oxidative phosphorylation, glycolysis, stemness, signaling, DNA repair, maintenance of genomic stability, etc. part or all of which are involved in tumor suppression, as has been extensively reviewed recently [24,53,54,56,67,68,69]. This also explains why *p53* mutations are associated with more than 50% of human malignancies, including aflatoxin-induced HCC [31].

### 4.4. Oxidative Stress-Mediated Genotoxicity

Although the mutagenicity of aflatoxins has been primarily attributed to the formation of aflatoxin-N^7^-gua DNA adducts discussed above, it is becoming increasingly evident that it also arises from the oxidative stress (OS) produced by AFB1 metabolism. The OS acts directly on the DNA to induce the so-called oxidative DNA damage (ODD) or indirectly via the formation of by-products from lipid peroxidation (LPO) of membrane phospholipids [18,70,71] (Figure 1). Processing of AFB1 in the liver by CYP450 enzymes induces OS, releasing excessive amounts of reactive oxygen species (ROS) that can attack nitrogen bases and deoxyribose moieties of the DNA and generate more than 100 different DNA adducts [18,72,73] (Figure 1). The most known and best studied of these adducts is 7,8-dihydro-8-oxo-2′-deoxyguanosine (8-hydroxydeoxyguanosine, 8-oxo-dG, 8-OH-dG) derived from the oxidation of the DNA guanine residue by the hydroxyl radical generated by the OS, which is commonly used as a biomarker for oxidative DNA damage [18,72,74]. Intraperitoneal injection of AFB1 to rats increased, in a dose- and time-dependent manner, the levels of 8-oxo-dG in the liver, which was prevented by a pre-treatment of rats with the antioxidants selenium and deferoxamine, thereby confirming the relationship between the adduct and the oxidative stress induced by the aflatoxin [75]. Likewise, intraperitoneal injection of a single tumorigenic dose of 50 mg/kg AFB1 to mice increased the levels of 8-oxo-dG by about three-fold in alveolar macrophages and non-ciliated bronchiolar cells (Clara or Club cells) preparations isolated from mice scarified 2 h after the treatment; no such increase was observed in liver tissues of the mice [76]. Consistent with these findings regarding the absence of the adduct in the liver, a recent study showed no significant increase in seven ROS-modified bases, including 8-oxo-dG, in the liver tissues of rats treated with 7.5 mg/kg AFB1, as compared with control rats (untreated); whereas the levels of 8,5′-cyclo-2′- deoxyadenosine, another DNA adduct from oxidative attack of the adenine base, increased significantly compared with background levels in control rats [32]. The extent of oxidative DNA damage, the type of adduct produced, and the efficiency and speed in DNA repair were reported to be dependent on the species, organ, tissue, sub-cellular component, and cell cycle [18]. The lung appears to be the most common target for DNA damage with 8-oxo-dG accumulating mainly in mitochondria and the nucleus [74,77,78]. A recent study demonstrated that AFG1 upregulated the expression of tumor necrosis factor (TNF)-α and CYP2A13 in mice alveolar type II (AT-II) cells of lung tissues, and in vitro in human AT-II-like cells (A549), which mediate an inflammation with increased numbers of γ-H2AX- and 8-OHdG-positive cells in the inflamed tissues [79]. According to the authors, the inflammatory reaction induced by TNF-α upregulates the expression of CYP2A13, which in turn sustains active metabolism of AFG1 leading to ODD, as evidenced by the increased expression of the DNA damage marker γ-H2AX. Like AFBO-derived DNA adducts, 8-oxo-dG lesions mediate G→T transversion mutations, but they do not specifically target the *p53* gene and they involve different mechanism and different DNA polymerases [75].

Regardless of the source of OS-induction, this is a frequent phenomenon in cells, which is normally counteracted by physiological mechanisms involving antioxidant systems consisting of enzymatic antioxidants, such as superoxide dismutase, catalase, and glutathione peroxidases, and thioredoxin, or antioxidants metabolites [18,80]. It also triggers modulatory signaling pathways to balance the associated inflammatory reactions, among which Nrf2 (Nuclear erythroid-2 related factor-2)/ARE pathway was reported to play a pivotal role. In this pathway, the Nrf2, a transcription factor that is normally sequestered by Keap 1 (Kelch-like ECH-associated protein) is liberated and upregulates the ARE gene expression of detoxification enzymes, such as haemoxygenease-1 (HO-1), which inhibits pro-inflammatory cytokines and activates anti-inflammatory cytokines [81]. Yet, the above-mentioned OS-modulatory means may be of limited efficacy to prevent DNA damage that should be repaired before replication in order to preserve the genomic stability and prevent cumulative mutations and genotoxicity [18]. The 8-oxo-dG lesions are primarily repaired by BER mechanisms using, for example, 8-oxoguanine DNA glycosylase 1 (OGG1) enzyme, a multifunctional DNA glycosylase that specifically recognizes the damaged base and excises it by breaking the N-glycosidic bond. The enzyme then cleaves the DNA backbone leaving an AP site to be filled by the appropriate nucleotide in subsequent steps using specialized DNA polymerases [71]. However, when the amounts of ROS are too high to be balanced by cellular antioxidant defense mechanisms, the DNA damages cannot be timely repaired and they accumulate to produce various genetic abnormalities, including erroneous gene expression, multiple mutations, genomic instability, and eventually tumorigenicity [18]. The production of 8-oxo-dG by OS has indeed been reported to be an important means by which AFB1 causes cancer in various organs in humans and animals [32,75,82].

As was mentioned above, AFB1 can also induce DNA damage by OS indirectly via the production of ROS which, in turn, attack oxidatively membrane phospholipids and release different mutagenic aldehydes (Figure 1). Among 33 known LPO-derived aldehydes, malondialdehyde (MDA), acrolein (Acr), 4-hydroxy-2-nonenal (HNE), and 4-oxo-2(E)-nonenal (ONE) are the most predominant and can react with DNA bases to generate pro-mutagenic exocyclic DNA adducts leading to genomic instability and possibly to carcinogenicity [83,84,85,86,87,88,89]. For example, acetaldehyde (Ace), Acr, Cro, and HNE can bind to the DNA guanine residue to from the highly mutagenic 1,N^2^-propano-2′-deoxyguanosine (1,N^2^-propanodGuo, 1,N^2^-PdG) adduct [70,88,90,91,92]. However, few studies, to our knowledge, have identified the specific aldehydes produced by AFB1-induced LPO and their contribution to cancer development in humans or animals. An early study demonstrated a dose-dependent production of MDA and conjugated dienes in rat liver homogenates treated with AFB1; and this production was prevented by a pre-treatment of the cells with antioxidants and iron chelators [93]. The study also demonstrated that the aldehydes produced accumulate in the cellular microsomes, nucleus, and mitochondria and damage them. A subsequent study conducted in the same laboratory further demonstrated that lipid peroxidation in hepatocytes increased with increasing doses of AFB1 of 10–100 mM [94]. Although the latter study provided evidence for the implication of hydrogen peroxide and hydroxyl radical as the main AFB1-generated ROS responsible for LPO, it provided no indication about the nature of the resulting aldehydes. Conversely, a recent study showed that AFB1-induced OS in human hepatocytes (HepG2) released Acet and Cro with a subsequent formation of the highly mutagenic cyclic α-methyl-γ-hydroxy-1, N^2^-propano-deoxyguanine (meth-OH-PdG) DNA adduct [95]. Interestingly, the study demonstrated that OS plays more prominent role in AFB1-mediated mutagenicity than does AFBO, as was substantiated by the following findings: (i) AFB1 treatment of the human hepatocytes generated more than 30 times higher levels of meth-OH-PdG adducts than AFB1-N^7^-gua, (ii) like AFB1, Acet and Cro targeted the hotspot codon 249 on *p53* gene to cause G→T transversion mutations, but they had higher preferences for the site than AFBO, and (iii) the DNA repair of the meth-OH-PdG lesions produced by Ace and Acr was significantly slower than that observed with AFBO-derived DNA lesions. Indeed, strong inhibition of both BER and NER by LPO-generated aldehydes and its enhancement by the accompanying epigenetic modifications is well documented [96,97,98]. In addition to reduced DNA repair of the meth-OH-PdG-mediated lesions, methylation of the cytosine on codon 248 (-*CGG-) was shown to promote the adduct formation on the adjacent codon 249 of the tumor suppressor *p53* gene [95]. Moreover, the concomitant production of 8-OH-dG by ODD, discussed above, may enhance the epigenetic effect, as this adduct was shown to increase cytosine methylation at the -*CpG- islands [99]. On the other hand, it is well established that reactive LPO-induced aldehydes are more mutagenic than the free radicals and the highly reactive AFBO [83]. Indeed, reactive aldehydes can act remotely from the site of their formation, contrary to the short-lived and highly instable free radicals and epoxides. This may also account for a reason why AFB1 can mediate cancer in organs distant from the liver where the aflatoxin is activated and metabolized [5]. Despite the shortage in studies related to aflatoxin-mediated OS, the available data clearly suggest that OS plays more important roles in aflatoxin toxicities than presently assumed. In fact, the above-mentioned study of Weng, Lee, Choi [95] suggests that the role of OS has been undermined so far and should be further investigated. The high efficacy of selenium in preventing HCC onset in chicken provides additional evidence for the implication of OS in the tumorigenicity of aflatoxins, since this trace mineral acts primarily by enhancing the anti-oxidative capacity of cells [100,101]. It also provides an additional hint for the toxicity of aflatoxins which do not involve the formation of the reactive epoxide upon metabolic activation by the liver cytochrome enzymes, such as AFB2 and AFG2 and other AFB1 metabolites [102] whose furan ring does not have the double bond between C^8^ and C^9^ carbons.

## 5. Immunotoxicity

Increased frequency and severity, and prolonged healing of infectious diseases, in addition to decreased vaccination efficacies provided evidence that aflatoxins disrupt both innate and acquired/adaptive immunity [103,104,105,106,107]. The general mechanisms of AFB1 immunotoxicity via AFBO is presented in Figure 1. It can be seen from this figure that AFBO interacts with immunocompetent cells throughout the body to affect their proliferation and/or production of immune response mediators, thereby disrupting the innate and adaptive immunity. Although most studies to illustrate these mechanisms have been carried out on animals, the immunotoxicity of AFB1 has also been substantiated in vitro on human cell lines and in case–control studies in highly exposed regions, e.g., Ghana [105,108,109,110]. However, few studies to our knowledge have investigated the immunotoxicity of aflatoxins other than AFB1 or its combination with other mycotoxins [111,112,113,114]. Meanwhile, there has been a general agreement that low or moderate concentrations of AFB1 have no or a marginal immunotoxicity, and that cell-mediated immunity (CMI) is more susceptible to aflatoxins than humoral immunity [105,112,115,116].

A concentration of 60 µg AFB1/kg feed given ad libitum to weanling pigs for 33 days had no noticeable effects on the counts of different types of leukocytes and lymphocytes, or on antibody and cytokine titres, while the highest concentration tested of 180 µg/kg feed had only moderate effects on leukocyte counts and TNF-α [113]. This appears to be especially relevant that young pigs are among the most susceptible hosts to aflatoxins [117]. In rats, the oral administration of 100 µg AFB1/kg bw once a week for five weeks only slightly inhibited the proliferation of lymphocytes, with no significant effect on the related secretions of cytokines, chemokines, and immunoglobulins in the serum; a ten-fold higher dose of 1 mg AFB1/kg bw could only increase numbers of CD8^+^ (cytotoxic T lymphocytes), while various other immunological parameters remained unchanged [118]. Similarly, feeding rats ad libitum on feed contaminated with AFB1 at different levels (0.01–1.6 mg/kg feed) for longer periods (up to 40 weeks every other 4 weeks) exerted significant effects on the immune response only at the highest concentrations of 0.4 and 1.6 mg/kg after 12 weeks of exposure or longer [119]. Nonetheless, other studies suggested that lower concentrations of aflatoxins and shorter durations of exposure can still alter the immune response. For example, feeding rats with diet containing 5 to 75 µg AFB1/kg bw for five weeks [120], or dosing mice intraperitoneally with 25 or 50 µg AFM1/kg bw five days per week for 4 weeks [111] have altered their immunity in a time- and dose-dependent manner.

On the other hand, the higher susceptibility of CMI compared with the humoral immunity is well documented, as has been reviewed previously [105,110]. For example, dosing rats with 0.6 mg AFB1/kg bw had no significant effect on IgM titre, while ten-fold lower dose (0.06 mg/kg bw) could inhibit lymphocyte proliferation [121]. Moreover, the ingestion of 0.1 or 1 mg AFB1/kg bw did not alter anti-ovalbumin IgE and IgG antibody production in rats mesenteric lymph nodes despite their significant action on the proliferative activity on B and/or T lymphocytes [118].

Regarding the mode of immunomodulation of the immune function by aflatoxins, most of the available data suggest that they mainly exert suppressive effects; however, in vitro and in vivo studies have demonstrated that they can also dysregulate the immune response via immunostimulatory effects [122,123].

### 5.1. Immunosuppression

Immunosuppression is manifested by the destruction of the physical barriers as a first line defense against invaders (pathogens and toxins), the inhibition of proliferation and function of immunocompetent cells, or the decrease in complement system activity, thereby interfering with both innate and adaptive immunity [105,110].

#### 5.1.1. Innate Immunity

The destruction of physical barriers such as the skin and the intestinal epithelial cells with a consequent impairment of the barrier function against microbial and toxin invasions has been demonstrated in vivo and in vitro. Contact of AFB1 with the skin of different animals was reported to elicit various types of lesions spanning from the formation of intra-epidermal vesicles to squamous cell carcinoma [114,124,125,126]. Feeding pigs for 28 days on feed contaminated with mixtures of aflatoxins (AFB1, AFB2, AFG1, and AFG2) produced crusting and skin ulceration on the snout, lips, and buccal commissures [114]. Aflatoxins have also been demonstrated to disrupt the integrity and function of the mechanical barrier of intestines by interfering with the cell cycle progression or by destroying the intestinal epithelial cells and the tight junctions (TJs) that cement them together. Administration of 0.6 mg AFB1/kg diet to broilers for 3 weeks stalled the cell cycle at the G2/M phase causing a reduction in the height jejunum and in the ratio of villus height/crypt, thereby impairing their function as a selective barrier [127]. These findings were recently corroborated by feeding broiler chicken with feed containing 0.6 mg AFB1/kg for up to 21 days and monitoring structural and functional changes in the small intestine [128]. The study showed various structural and histopathological injuries similar to those described above regarding the increased depth of villi with decreased height and area [127], in addition to other histopathological alterations in the small intestine, including mitochondrial vacuolation and loss of cristae, reduced numbers of the absorptive cell goblets and the junctional complexes. Such changes dramatically alter the barrier function of the intestine to interfere not only with nutrient absorption, but also with the innate immune response as a protective means against the invasion of pathogens or toxins. Indeed, increased gut permeability was induced in broilers fed on feed contaminated with 1.5 mg AFB1/kg for 20 days [129]. Lower concentrations of aflatoxins AFB1 (16.3–134 µg/kg feed) and AFB2 (3.15–23.6 µg/kg feed) orally administered to broilers for up to 42 days disturbed the cell cycle progression and apoptosis causing histopathological lesions with different severities in thymus and bursa fabricius where T and B lymphocytes undergo maturation, respectively [130]. At the molecular level, aflatoxins have been demonstrated to alter the mechanical, chemical, and immune barriers that protect the intestinal mucosa against various external threats. In vitro exposure of human cell line CacO-2 to 1–100 µM of AFB1 for 48 h decreased the trans-epithelial electrical resistance (TEER) with consequent increase in the paracellular permeability and decrease in the viability [131]. The study related the latter effects to downregulation of the transcription of three constitutive proteins of the TJs, claudin-3, claudin-4, and occludin. In a similar study, CacO-2/TC7 cells exposed to AFM1 (3.2 and 33 nM) for 24 h showed reduced TEER of the monolayer and accelerated transport of the aflatoxin through it, meanwhile, the TJs and their constitutive proteins remained intact [132]. Likewise, the selective permeability of CacO-2 cells was disrupted upon exposure to different amounts of AFM1 (0.2 to 20 µM) for 48 h [133]. The latter study associated the permeability disruption to reduction of TEER, down-regulation of the expression of structural TJ proteins (claudin-3, claudin-4, occludin, zonula occludens-1), and decrease in the levels of p44/42 mitogen-activated protein kinase (MAPK) involved in cell death or cell survival. Other AFB1-induced structural disturbances of the gastro-intestinal tract that alter immune functions with special focus on broiler chicken have been thoroughly reviewed previously [123].

Effects of aflatoxins on immune cells that play key roles in the innate immunity, such as monocytes, macrophages, dendritic cells (DC), and natural killer (NK) cells to restrict their viability, function, or genetic expression of cytokines and chemokines is well documented (Figure 1). Exposure of broilers to AFB1 was reported to repress the transcription of toll-like receptors (TLR) TLR-2, TLR-4, and TLR-7, indicating a suppressive effect on the innate immunity where these receptor proteins are involved in the recognition of external invaders by sentinel cells, e.g., macrophages and dendritic cells, as a key step to trigger this type of immune response [128]. AFB1 at the low dose of 10 ng/mL was also reported to reduce the antigen-presenting activity of porcine dendritic cells, although this reduction could not be associated with down-regulation of the expression of TLRs or specific cytokines [134]. Moreover, aflatoxins AFB1, AFB2, and/or AFM1 were reported to reduce viability, proliferation, cytotoxicity, and phagocytic activity of macrophages as well as their expression of cytokines, e.g., TNF-α, IL-1, and IL-6, and the inducible nitric oxide synthase (iNOS) that mediate intracellular killing of pathogens in phagocytosis [112,135,136,137,138,139]. Recently, AFB1 was demonstrated to dysregulate the innate immune function mediated by autophagy and external trap formation in M1-type macrophages responsible for inflammatory reaction, which is triggered by the secretion of pro-inflammatory cytokines, such as TNF-α, IL-1β, IL-6, IL-12, IL-23, [122]. Pre-treatment of human monocytes with as low concentration as 0.1 pg AFB1/mL for 24 h prior to incubation with *Candida albicans* for 30 min at 37 °C has impaired significantly their phagocytic and killing activities towards the pathogenic yeast [140]. The in vitro reduction of chemotactic response to bacterial chemoattractant factor, a phagocytic stimulatory mediator, was demonstrated on neutrophils harvested from the blood of piglets that had been suckling milk contaminated with AFB1, AFM1, and AFG1 [141]. Intraperitoneal administration of AFM1 to mice at doses of 25 and 50 mg/kg bw reduced significantly its phagocytic activity against *E. coli* [111]. As regards the effects of aflatoxins on the proliferation and cytotoxic activity of NK cells, conflicting data are available in the literature. While mice gavage with 0.03–0.7 mg AFB1/kg inhibited cytolysis of YAC-1 cells by NK cells in BALB/c [142], the same concentrations or even higher (24 mg/kg bw) had no such an effect in C57B1/6 mice [143]. However, a significant reduction in the proliferative and cytotoxic activities of human NK cells was demonstrated in vitro upon incubation of the cells with 0.005–0.05 ng AFB1/mL [144]. Phagocytic and cytotoxic activities of dairy cow neutrophils against *Staphylococcus aureus* and *Escherichia coli* were also dramatically hampered upon exposure to low doses of AFB1 (0.01, 0.05 and 0.5 ng/mL) for 18 h, which was ascribed to the depletion of neutrophil cytosol from ROS, playing pivotal role in the killing process of pathogens during phagocytosis, rather than affecting the viability of the neutrophils themselves [145].

The complement system—as a crucial component of the innate immunity that activates phagocytosis of infectious pathogens—was shown to be inhibited by aflatoxins in various animals. Dosing guinea pigs per os daily with 30 µg AFB1 or greater amounts for 20 days decreased the complement activity [146], while a dose of 10 µg had no noticeable effect on these innate immune mediators [147]. A decrease in the complement activity was also demonstrated by feeding trials in cattle and poultry at different threshold levels [115,123,148]. Concentrations ranging between 0.11 to 0.21 mg AFB1/kg feed were shown to impair both the classical complement pathway and the alternative pathway of complement activation (APCA) in ducklings [149]. However, according to Valtchev, Koynarski, Sotirov [116], feeding ducklings with AFB1 at doses of 0.5 or 0.8 mg/kg feed for 40 days had a stimulatory effect on the APCA in the first 15 days, followed by suppressive effect during the next days of the experiment. Yet, the effect of aflatoxins on the complement system may depend largely on the host, as no significant change in the serum hemolytic activity (CH_50_) was recorded in rabbits exposed to as high level as 24 mg/kg feed for 28 days [150]. AFM1 was demonstrated to reduce significantly the complement system in Balb/c mice receiving a dose of 25 or 50 μg/kg bw, as evidenced by a decrease in CH_50_ using rabbit anti-sheep red blood cells (RBC) IgG antibodies and sheep RBC [111].

#### 5.1.2. Adaptive Immunity

Suppression of adaptive/acquired immunity upon exposure to aflatoxins is well established indicating the increased vulnerability of exposed hosts to infectious agents, as well as the reduced or failed protection of vaccination [107,151,152]. The latter effect has been demonstrated in poultry by epidemiological studies correlating aflatoxin exposure to poor protection by vaccination against Newcastle disease [123] and infectious bronchitis [153]. Similar suppressive effects of immunization were reported in pigs, where vaccination failed to protect them against *Erysipelothrix rhusiopathiae* when given AFB1-contaminated feed, contrary to a control group receiving aflatoxin-free feeds [152]. Furthermore, decreased proliferation, activation, and/or function of lymphocytes, as the main immune cells that promote adaptive immunity, has been demonstrated in humans and animals. A dose- and time-dependent apoptotic effect was observed on human peripheral blood lymphocytes incubated with different doses of AFG1 (3.12–2000 µg/L) for different times (2–72 h) [154,155]. In vitro exposure of human lymphocytes to AFB1 at concentrations of 5–165 µM induced a dose-dependent increase in numbers of apoptotic and necrotic lymphocytes, with an evident rise in cell necrosis starting from 50 µM (~15.6 mg/L) at 24 h of incubation [156]. In vitro exposure of human lymphoblastoid Jurkat T-cell line to AFB1 or AFM1 at 3–50 µM for up to 72 h inhibited the proliferation of the T cells in a dose-dependent manner starting at 15 µM, but did not cause their apoptosis or necrosis [157]. According to the same study, AFB1 and AFM1 increased significantly the expression of IL-8 involved in innate immunity, while the adaptive immunity remained unaffected as suggested by unchanged levels of interferon (INF)-γ and IL-2 cytokine compared to negative control cells incubated in the absence of aflatoxins. A concentration of 10 mg AFB1/L or greater inhibited the differentiation of mitogen-induced T and B lymphocytes in cattle with a consequent impairment of both T-cell dependent and T-cell independent humoral immunity, and hence immunoglobulin production [158]. Up-to 10 mg AFB1/kg feed was required to suppress IgG and IgA production by B lymphocytes and to restrict the humoral response against *Salmonella* and rabbit red blood cells in chicken [151,153,159]. Furthermore, intraperitoneal administration of 50 µg AFM1/L for four weeks (five times a week) to mice did not affect the concentration of IgM in the blood serum [111]. In addition, a dose of 1.8 mg AFB1/kg feed given to pigs for 18 days did not stimulate anti-ovalbumin IgG production in the serum despite the induction of mitogenic activity of lymphocytes, indicating that this dose of the aflatoxin specifically suppresses the activation of B lymphocytes but not their proliferation [106]. Concentrations below 0.5 mg AFB1/kg feed did not affect antibody responses to *Pasteurella multocida*, *Salmonella pullorum* and Newcastle disease in broiler chicken and turkey [115]. Although it is now well established that disruption of humoral immunity requires higher aflatoxin dosage than does CMI, no consensual threshold levels that alter CMI or humoral immunity have been reached so far. Such levels vary widely depending on the species, the age, the gender, and the route of administration. In poultry, doses of 0.4 and 1.0 mg AFB1/kg feed are the most accepted such thresholds for CMI and humoral immunity, respectively [123].

Suppression of adaptive CMI has been studied on laboratory animals, mainly poultry and rodents, demonstrating a decrease in numbers of different subsets of T-cell lymphocytes and cytokines they produce, as key elements in this type of immune response [123,160]. Adaptive CMI suppression by aflatoxins was also evidenced by decreased delayed-type hypersensitivity (DTH) in different animals at concentrations ranging between 0.3 and 1.0 mg/kg feed [123,161]. DTH was significantly delayed/decreased in broilers and turkeys receiving an AFB1 dose of 0.2 mg/kg feed or higher for 33 days [159]. Conversely, a subsequent study showed that a two-fold higher dose did not affect DHT in broilers, which was, by contrast, significantly reduced when the same dosage consisted of a combination of AFB1 and AFB2 [162]. A one-day-old broilers receiving 0.6 mg AFB1/kg feed for three weeks displayed reduced proportions of CD3^+^, CD3^+^CD4^+^, and CD3^+^CD8^+^ T-cell subsets as well as the transcription of different cytokines in the birds intestines, thereby impeding adaptive CMI [160], where these T-cell subsets and some of the inhibited interleukins, e.g., IL-2 and INF-γ, play a crucial role [163]. Proliferation and cytokine production by splenic helper T lymphocytes (CD4^+^) involved in acquired cellular immunity were also reduced in rats given AFB1 doses ranging between 5 and 75 µg/kg bw for five weeks [120]. Similar effects were induced by AFM1 in mice dosed intraperitoneally with at 25 or 50 µg/kg bw, 5 days per week for 4 weeks, where suppression of acquired CMI was evidenced by a decrease in DTH and related T lymphocytes subsets (CD3^+^, CD4^+^, CD8^+^, CD19^+^ and CD49b) as well as the interleukins they produce, e.g., INF-δ, IL-10, and IL-4 [111]. In humans, elevated levels of AFB1, as estimated by the concentrations of AFB1-albumin adduct in the serum, were highly correlated with a decrease in lymphocyte subsets CD3^+^ and CD19^+^ bearing the D69 activation marker (i.e., CD3^+^CD69^+^ and CD19^+^CD69^+^), and CD8^+^ T-cells which play a central role in vaccination and immune response against pathogens [109]. The latter results suggest that AFB1 impairs acquired CMI in humans and decreases their resistance to infections, consistent with the reported accentuation of impaired activation of CD8^+^ and CD4^+^ T lymphocytes in human immunodeficiency virus (HIV)-positive Ghanaian patients [164].

It should be pointed out, however, that humoral immunity and CMI, whether they are acquired or innate, cannot always be separated. For example, any dysregulation of the proliferation and/or TLRs expression in dendritic cells will have direct repercussions on innate and adaptive immunity, as these antigen-presenting cells are key intermediates between both types of immune response [139,165].

### 5.2. Immunostimulation

Regarding the immunostimulatory effects, there is increasing evidence that aflatoxins illicit a biphasic immune response with a stimulatory action in the first phase and suppressive action in the second [123]. According to Valtchev, Koynarski, Sotirov [116], exposure to low doses of aflatoxins for short periods stimulates the immune system, while exposure to higher doses for longer periods exerts immunosuppressive effects. For example, the transcription of TLR-2 and TLR-4 was upregulated in human myeloid dendritic cells (DC) exposed to environmentally relevant doses of AFB1 (1 or 2 µg/L) for 2 to 24 h [108]. The upregulation of TLR expression has been demonstrated in different immune cells from different organs as a means of sensing very low levels of aflatoxins [21,108,139,145]. A single dose of 663 μg AFB1/kg bw given to mice by gavage upregulated the production of both the inflammatory cytokine IFN-γ and the anti-inflammatory cytokine IL-4 after 5 days of ingestion [166]. The authors attributed such an unusual reaction to the activation of innate immune cells after a short time of administration of a high dose in a single shot, as a first step preceding the trigger of an adaptive response. Intermittent intake of AFB1 simulating the actual situation was also reported to result in an alternation of suppressive and stimulatory/compensatory effects upon exposure and resting (aflatoxin-free diet) periods, respectively [119]. Despite the conflicting data and lack of consensus regarding the cytokine types induced in response to aflatoxin exposure, the unnecessary upregulation of the immune response stimulates the production of tissue-damaging inflammatory molecules and free radicals leading to chronic inflammation, cancer, and nervous system degenerative diseases [21,79]. Moreover, low levels of a mixture of aflatoxins (AFB1, AFB2, AFG1, and AFG2) increased the antigen-presenting capacity of dendritic cells that stimulate T-cell proliferation, which has been suggested to breakdown the immunological tolerance and increase host susceptibility [134].

## 6. Teratogenicity

The exposure of pregnant females or birds to aflatoxins can affect embryos in utero or in fertilized eggs, respectively, producing various adverse health effects and different pathological gestation/incubation outcomes [167]. In mammalians, systemic blood circulation in highly exposed mothers conveys aflatoxins or their toxic metabolites to foetuses, as has been substantiated in highly exposed pregnant women from African and Asian countries, as well as in animals. Indeed, aflatoxins and/or biomarkers derived thereof, e.g., aflatoxin metabolites, and aflatoxin-DNA and aflatoxin–albumin adducts, were detected in the cord blood of the foetus or in both foetal cord and maternal blood samples [168,169,170,171,172]. Accordingly, it was concluded that aflatoxins or their metabolites in pregnant women are transmitted to the foetus and metabolized through the same pathways as in adults [172]. Therefore, the pregnancy of highly exposed mothers is prone to various outcomes, including foetal growth restriction, foetal loss, and premature birth. Growth restriction has been documented in humans and animals where an inverse relationship between the birthweight and the amounts of appropriate biomarkers in the cord blood has been extensively demonstrated [169,173,174,175]. Conversely, few studies have related high-aflatoxin exposure of pregnant women to stillbirth, while studies on the association of high aflatoxin intake by pregnant women with premature birth and foetal loss are either non conclusive [175] or lacking [174]. On the contrary, decrease in live birth and litter size, impairment of organ development, and skeletal anomalies in offspring have been demonstrated in animals given aflatoxins at daily doses ranging between 0 (nil) and 100 µg/kg bw, which was explained by the binding of aflatoxins to the DNA and the hindrance of protein synthesis [167,176,177,178,179,180]. This view can be applied to humans, as aflatoxins bind to human DNA in the same way, but it remains to be clinically demonstrated.

In addition to the above-mentioned adverse health effects, an aflatoxin-rich diet in pregnant females affects their health and expose their foetuses to indirect consequences with congenital abnormalities. For example, upregulation of maternal pro-inflammatory cytokines and/or downregulation of anti-inflammatory cytokines induce systemic inflammation that impairs the placental growth and causes its insufficiency ultimately leading to poor foetal growth, miscarriage and stillbirth, or prematurity [113,168,173]. Furthermore, the cytotoxic activity of aflatoxins induces anaemia in mothers by lysing red blood cells or interfering with nutrients, e.g., iron, selenium, and vitamins, absorption with consequent poor foetal growth and/or prematurity [181,182,183]. The association of anaemia and high aflatoxin intake, as determined by AFB-albumin adduct in the mothers’ serum, was demonstrated in a cross-sectional study on Ghanaian women [175]. On the other hand, the association of anaemia to red blood cell lysis by aflatoxins was demonstrated in vitro and in animal species dosed with 0.5 to 1.0 mg/kg bw [184,185,186,187]. However, it appears that the environmentally relevant levels of aflatoxins remain below the doses that can elicit red blood cell lysis in humans. Conversely, there is a lack of evidence on the association between inflammation-induced anaemia in pregnant women and their exposure to aflatoxins. As matter of fact, there are many gaps in the knowledge of doses, mechanisms, and outcomes of exposure to aflatoxins in pregnant women that require more attention and rigorous scientific approaches to be clearly understood and eventually avoided to ensure safe pregnancy and birth.

## 7. Other Adverse Health Effects of Chronic Exposure to Aflatoxins

In addition to the major toxicological effects reviewed above, aflatoxins exert various other adverse health conditions with overlapping mechanisms and risk factors. These include malnutrition diseases (faltering and stunting), retarded physical and mental maturity, reproduction and sexuality issues OK, and nervous system diseases (neurodegenerative diseases and neuroblastoma) [2,188,189,190]. However, most of the latter effects have been scarcely investigated to cover the main pertaining aspects from applied and mechanistic standpoints. Therefore, further studies are needed for clearer insights on these issues to have an accurate and realistic opinion on the risk they may pose to the public health. This section addresses malnutrition and neurodegenerative diseases, which have been relatively well studied.

### 7.1. Aflatoxins and Malnutrition

Malnutrition is probably one of the above-mentioned aspects that has received the most attention due to its impact on childhood in many developing countries, where children are already facing food shortages. It is essential to ensure children receive balanced and nutrition for healthy growth, and hence be well prepared to adulthood as active and productive individuals. Exposure to aflatoxins exacerbates such poor nutritional status by interfering with the absorption of vitamins and minerals, as has been shown for vitamins A, C, and E, and selenium [191]. This not only deprives children/consumers from these essential micronutrients, but also increases their susceptibility to aflatoxins that they normally detoxify owing to their inherent antioxidant or CYPP450 inhibitory activities [101,167,192]. As a result, exposed children may experience growth disorders from the gestational stage as discussed above, throughout adulthood, with stunted and retarded physical and mental maturity [193]. Indeed, in African countries, growth faltering among children below 5 years old was correlated with chronic exposure to high levels of aflatoxins when they rely on local agricultural products, e.g., maize, peanut, and derivatives as staple foods [194]. On the other hand, severe protein energy malnutrition (PEM) diseases, such as Kwashiorkor and marasmic kwashiorkor, have been associated with chronic exposure to high levels of dietary aflatoxins in different African countries [195,196,197,198]. However, since all the relevant studies were conducted in poor household environments where children were invariably fed on local agricultural products with poor nutritional and hygienic quality and limited availability, PEM could be due to the limited access to enough nutritious foods, rather than to aflatoxin intake. To address this particular issue, a study has been conducted on malnourished Soudanese children with Kwashiorkor, marasmic kwashiorkor, or marasmus. The results of the study revealed that a group of kwashiorkor and marasmic kwashiorkor children had significantly higher levels of AFB1 and its derivative aflatoxicol in their sera and urine compared with a group of malnourished children with marasmus and a group of age-matched normally nourished children [197]. Accordingly, the authors concluded that kwashiorkor is definitely correlated with high chronic exposure to aflatoxins as either secondary to liver damage or an aetiological factor of the disease, which remains to be further substantiated by appropriately designed future studies [197].

### 7.2. Aflatoxins and Neurodegenerative Diseases

In addition to the classically known adverse health effects of aflatoxins, there is increasing body of evidence that chronic exposure to aflatoxins can also be responsible for neurodegenerative disorders. The AFBO and ROS generated by CYP450 enzymes and aflatoxin-induced oxidative stress, respectively, react with functional macromolecules in neuronal brain cells where they inhibit lipid and protein synthesis to induce their degeneration [199]. Aflatoxins were also reported to disrupt the structure and function of mitochondria of brain cells, which impedes oxidative phosphorylation and leads to their apoptosis [200]. In addition, the detection of aflatoxins in brain tissues of kwashiorkor-deceased children and their association with Rey’s disease (cerebral edema and neuronal degeneration) is a strong indication that aflatoxins can cross the brain-blood barrier and infiltrate the nervous system that they degenerate [21,188]. Although scarce, epidemiological studies have demonstrated the neurotoxicity of aflatoxins in humans and animals. In a recent study, rats dosed with 1/600th their LD_50_ dysregulated the levels of biochemical biomarkers of the oxidative stress indicative of neurodegenerative disorders, which were corroborated by histopathological and immunohistochemical tests showing vasodilation, necrosis and astrocytes gliosis [188]. In addition to the oxidative stress, aflatoxins induce neurodegenerative disorders by dysregulating the immune response of immunocompetent cells and creating proinflammatory conditions in the central nervous system [21].

## 8. Acute Toxicity

The mechanism of acute aflatoxicosis is poorly understood, although many authors refer to the interaction between aflatoxins and macromolecules (proteins, phospholipids, and nucleic acids) with a consequent formation of various adducts, which in turn interferes with the physiological and structural functions of the macromolecules. In particular, aflatoxin-protein adducts have been the most frequently associated with acute intoxication, as this blocks protein synthesis, especially the enzymes involved in vital functions, such metabolic pathways, protein synthesis, DNA replication and repair, and immune response (Figure 1). Additionally, there is increasing evidence that aflatoxin-phospholipid adducts and ROS-induced LPO are the main reasons for the disruption of the integrity and function of the membranes of the cells, mitochondria, and endoplasmic reticulum [14,20], as depicted in Figure 2. Moreover, severe DNA fragmentation upon exposure to high doses of aflatoxins is another major effect of acute aflatoxicosis (Figure 1), as was observed in the testicular tissues of mice injected with a daily dose of 20 µg AFB1/kg bw for 21 days [62]. However, a recent study on the acute toxicity of AFB1 in poultry suggested that aflatoxin–dihydrodiol (AF–dhd) is the main metabolite responsible for acute aflatoxicosis for being the pivotal metabolite leading to the formation of aflatoxin–albumin adducts [117]. According to the authors, AF–dhd derives from aflatoxin-exo 8,9-epoxide and forms the aflatoxin–albumin adducts via aflatoxin-aldehyde bypassing the formation of aflatoxin–dialcohol of the detoxification pathway [13]; and the more rapidly and abundantly AF–dhd is formed, the higher is the mortality rate. Moreover, the metabolism of AFB2_a_ as a dietary contaminant or as an AFB1-phase I metabolite was also suggested to be involved in acute toxicity; apart from the formation of aflatoxin–albumin adducts, AFB2_a_ was also reported to bind covalently with cellular proteins and phospholipids, yielding lipid- and protein-adducts, possibly leading to acute aflatoxicosis [14].

It should be pointed out, however, that chronic exposure to low doses of aflatoxins can produce similar effects as those observed in acute aflatoxicosis; however, their effects can be mitigated by detoxifying phase II enzymes and cellular antioxidant defense mechanisms, or by DNA repair to prevent mutations, as discussed above (Section 4.). Alternatively, these effects accumulate progressively with continuous exposure to low doses to, ultimately, evolve into liver cancer as the typical outcome of chronic exposure. Therefore, acute aflatoxicosis may result from an abrupt accentuation of most or all of the above-mentioned damages in a short time when the dose is too high. Although such a high dose remains to be specified, an overwhelming amount of aflatoxins can overcome the detoxifying capacity of the cell and drive the metabolism of the toxins towards the production of toxic metabolites causing severe DNA damage, the disruption of cell cycle progression, DNA fragmentation, metabolic disorders, cytotoxicity, and tissue necrosis, eventually leading to organ failure (Figure 1) in a short period. This may hold especially true as the adverse aflatoxin effects are cumulative [201,202]. For example, FAPy-DNA adduct burden that triggers tumorigenesis in rats was estimated to be one adduct per 250,000 nucleotides, i.e., 40,000 adducts/cell [203], which can either accumulate progressively with chronic exposure, or be reached in a short time in the case of exposure to abnormally high doses of AFB1.

## 9. Conclusions

Aflatoxins are widespread, highly toxic contaminants that require further research to clarify many essential aspects for better knowledge of their toxicity patterns and occurrence in foods and feeds, in order to adequately address their adverse effects on public health and economy. The information reviewed herein reflects the scarcity or lack of information on aflatoxins other the major ones (AFB1, AFB2, AFG1, and AFG2), whose occurrence in foods and feeds—and roles in toxicity—have so far been overlooked. In addition, despite the intensive work that has been carried out on the toxicity mechanisms of aflatoxins for more than five decades, it is clear that the extent and nature of health disorders are not well understood due to their high complexity and the intricate and overlapping risk factors, some of which may be confounding factors. However, the advent of new and sensitive analytical tools would allow for rapid progress in the elucidation of toxicity mechanisms of aflatoxins and help design new preventive or therapeutic means.

## Figures and Tables

**Figure 1 ijerph-17-00423-f001:**
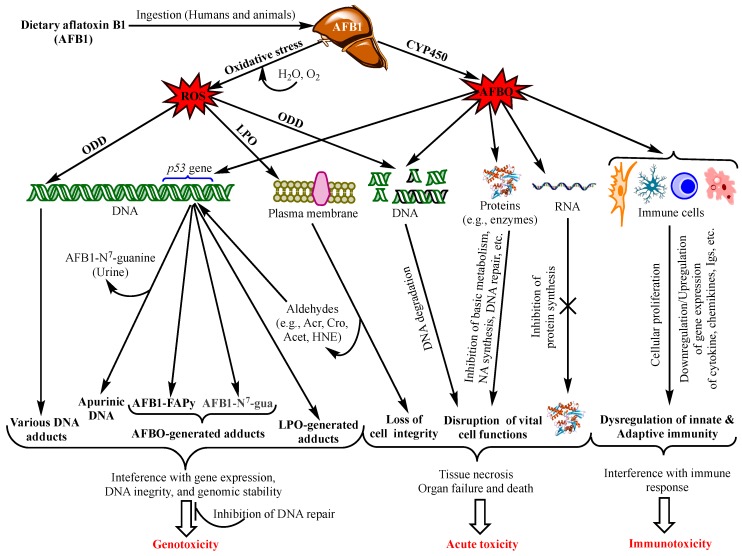
Main aflatoxin B1 toxicity mechanisms mediated by the oxidative stress and AFB1-exo-8,9 epoxide (AFBO; see text for explanations). NB: ROS also affect proteins, RNA molecules, and immunity as does AFBO (not shown in the figure. For details, see [20]). *Abbreviations*: AFBO: Aflatoxin B1-exo-8,9-epoxide; NA: Nucleic Acids; ROS: Reactive Oxygen Species; LPO: Lipid Peroxidation; ODD: Oxidative DNA Damage; Acr: Acrolein; Cro: Crotonaldehyde; Acet: Acetaldehyde; HNE: 4-Hydroxy-2-Nonenal; uFA: Unsaturated Fatty Acids; IL1β: Interleukin 1β, IL6: Interleukin 6; TNFα: Tumour Necrotizing Factor α; P-dG: Cyclic Propano-Deoxyguanosine; Igs: Immunoglobulins. See text for the other abbreviations.

**Figure 2 ijerph-17-00423-f002:**
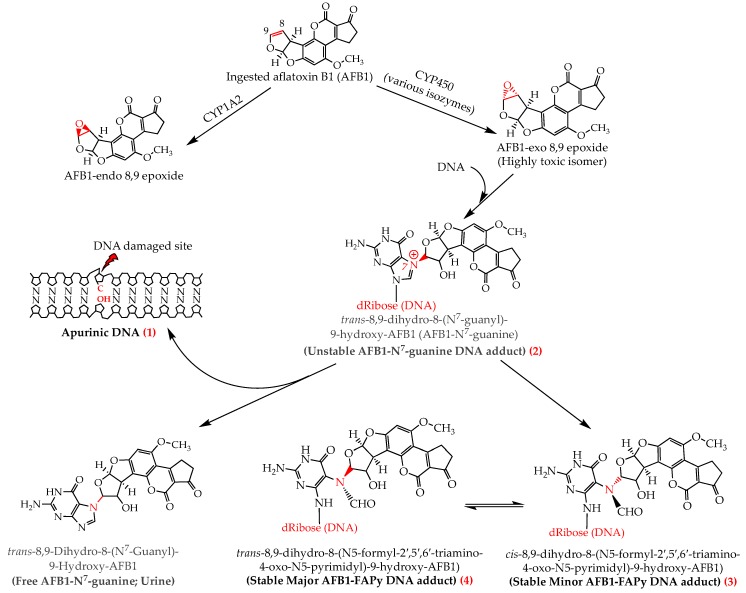
Activation of aflatoxin B1 and its interaction with the DNA leading to the formation of aflatoxin DNA adducts which cause three main DNA lesions, AFB1-N^7^-guanine (**1**), apurinic DNA (**2**), and AFB1-FAPy (**3**,**4**), involved in mutagenicity and carcinogenicity. Upon furan ring opening to stabilize the AFB1-N^7^-gua DNA adduct, the “*cis*” (minor) rotamer (**3**) of AFB1-FAPy is formed first and is then transformed into the “*trans*” (major) rotamer (**4**) to an equilibrium where the major rotamer is predominating (2:1; major to minor ratio) [52].

**Figure 3 ijerph-17-00423-f003:**
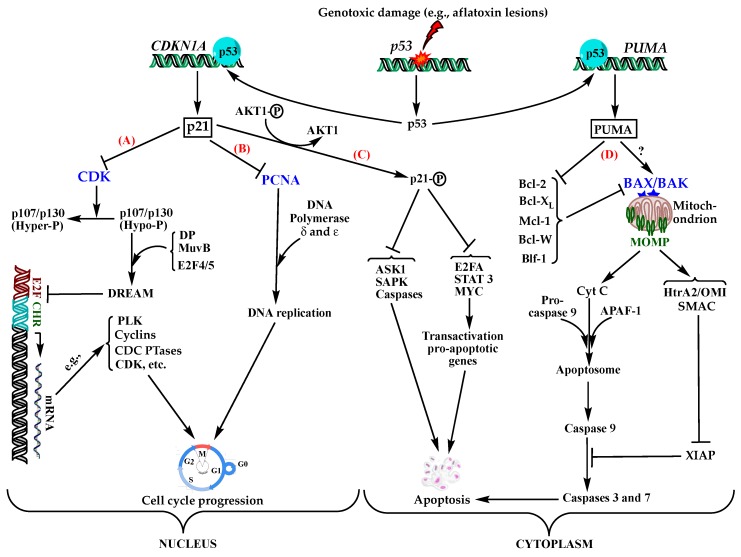
Main mechanisms used in normally functioning cells to induce cell cycle arrest or apoptosis as a response to DNA damage affecting *p53* gene to inhibit cell cycle progression in the nucleus (**A**,**B**), or apoptosis in the cytoplasm (**C**,**D**). (**A**) p21, as a potent inhibitor of CDKs, inhibits the phosphorylation of p107 and p130 proteins, which in their hypo-phosphorylated states can bind to MuvB core complex, E2F4-5, and DP and form an active DREAM complex. Once formed, DREAM binds to E2F and CHR promoters and represses the transcription of many genes, e.g., polo-like kinases (PLK1), cyclins A, B1, and B2, CDK1, CDCs 20, 25A, and 25C, MCM5, BIRC5, etc., involved in the progress of the cell cycle at different stages and checkpoints, thereby arresting the cell cycle at any stage of the progression depending on the gene(s) inhibited [55]. In the absence of p21, CDKs remain active and hyper-phosphorylate p107 and p130 preventing them from binding to the other DREAM components, thereby leaving E2F and CHR promoter sites free to bind transcriptional activators that, on the contrary, promote the cell cycle progression [54,56]. (**B**) p21 interacts with PCNA in the nucleus and prevents it from binding to the δ subunit of DNA-polymerase, which blocks DNA replication as well as DNA repair, among other functions ensuring the fidelity of DNA duplication [57]. (**C**) p21 can be phosphorylated by the serine threonine kinase AKT1 and prevented from translocating into the nucleus; in the cytoplasm, it acts as an anti-apoptotic factor that inhibits pro-apoptotic enzymes, such as ASK, SAPK, and different caspases. It also inhibits transcriptional factors, such as E2F1, STAT3, and MYC preventing the transactivation of pro-apoptotic genes [54,58,59]. (**D**) p53 transactivates *PUMA* gene as the major p-53-dependent mechanism for intrinsic apoptosis induction. Under normal conditions and in the absence of stimuli, apoptosis is restricted by five pro-survival proteins of the Bcl-2 family; Bcl-2, Bcl-X_L_, Mcl-1, Bcl-W, and Blf-1. Upon exposure to genotoxic stimuli, such as aflatoxins, p53 upregulates the expression of PUMA, a member of the Bcl-2 homology 3 (BH3)-only family, which inhibits all of the five pro-survival Bcl-2 proteins, thereby de-repressing the pro-apoptotic proteins BAX and/or BAK. This initiates mitochondrial damage allowing leakage of pro-apoptotic proteins through MOMP formation upon oligomerization of BAX/BAK, namely cytochrome C, HtrA2/OMI, and SMAC, which cooperatively induce apoptosis; cytochrome C binds APAF-1 and procaspase 9 to form an apoptosome and activate caspase 9 triggering the caspase cascade directly involved in apoptosis. Yet, caspase cascade can still be blocked by the pro-survival protein XIAP inhibitory to caspases 9 and 3. To proceed with apoptosis, SMAC and HtrA2/OMI combine to inhibit XIAP and relieve the caspases [60]?: In the absence or saturation of pro-survival Bcl-2 proteins, PUMA can directly activate BAX/BAK to resume the apoptosis process starting from MOMP formation, but this needs further studies to be ascertained [61]. Abbreviations: Bcl-2: B cell lymphoma-2; BH3-only: Bcl-2 homologue 3-only; Bcl-X_L_: B cell lymphoma extra-large; MuvB: Multivulval class B; DP: Dimerization partner; DREAM: Dimerization partner, RB-like, E2F and multivulval class B; CHR: Cell cycle gene homology region; PLK: Polo-like kinase; CDK: Cyclin dependent kinase; CDC: Cell division cycle; MCM: Minichromosome maintenance; BIRC: Baculoviral inhibitor of apoptosis repeat-containing 5; PCNA: Proliferating cell nuclear antigen; ASK: Apoptosis signal-regulating kinase; SAPK: Stress-activated protein kinase; STAT3: Signal transducer and activator of transcription; MYC: Myelocytomatosis; PUMA: p53-upregulated modulatory apoptosis; Mcl-1: Myeloid cell leukaemia-1; Blf: BCL-2-related protein isolated from foetal liver; BAX: Bcl-2-associated X protein; BAK: Bcl-2 antagonist/killer; MOMP: Mitochondrial outer membrane permeabilization; Cyt C: Cytochrome C; APAF-1: Apoptotic protease-activating factor 1; SMAC: Second mitochondria-derived activator of caspases; XIAP: X-linked inhibitor of apoptosis protein; HtrA2/OMI: High-temperature requirement protein A2; Hypo-P: Hypo-phosphorylated; Hyper-P: Hyper-phosphorylated.

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
