# Peer review of "Chronic and Acute Toxicities of Aflatoxins: Mechanisms of Action"

_ijerph, 2020, doi:10.3390/ijerph17020423_

Round 1

Reviewer 1 Report

The author wrote a respectable review article on aflatoxins, arguably the most complete to date.

It is more extensive than conventional reviews but it does not seem to be a problem because the the guidelines does not limit the number of pages.

In terms of content fluency, it was quite well-written, but there are some structural issues, overlooked journal guidelines and tips to be followed with potential to improve the work:

The abstract is too extensive. According to the guidelines, it "should be a total of about 200 words maximum";

There are plenty of blank spaces before and after the tables. Eliminate them, as they compromise the esthetic. It is not a problem to bring up as much text as possible and fill these gaps. There are exceptional cases when you use "landscape" pages, but in regular situations avoid blank spaces;

Some tables are too large. Some information could be sent as supplementary material. Most information is important for reading or studying but some is just relevant for consultation, confirmation or curiosity purposes. For instance, some aflatoxins are not as concerning as the well-known types B, G and M. You could send the remaining as supplementary material and refer in the caption or table footnote;

Keep figures and captions on the same page (e.g. figure 1). You can manipulate the spaces between the images or their sizes to achieve it;

Table 3: you should mention Mozambique because the country is historically relevant for aflatoxin research. It was there where Van Rensburg [1] conducted a pioneering study on the epidemiological association between aflatoxin intake and hepatocellular carcinoma, and later also van Wyk [2] presented an important article raising awareness on the need to reduce aflatoxins in groundnuts from Southern Africa in order to be able to export them to Europe. In Southern Africa, the latter study represent the turning point in aflatoxin research, from mostly a public health issue to an agricultural concern. The third turning point was certainly the 2004 outbreak in Kenya properly mentioned in your study. Considering the broadness of your review's scope, these studies seem important to mention to provide some context. Furthermore, there are more recent publications on aflatoxins in Mozambique.

Avoid as much as you can writing URL in the text as replacement to citations. For instance, you did that in table 2's caption. Write the full reference in the reference list and cite it in text just like any other source; Revise the page numbering;

Figure 4's caption is too extensive. Consider putting some of its content in the main text when discussing about the figure. Fit the caption within one page.

REFERENCE

1. Van Rensburg, S.J.; Cook-Mozaffari, P.; Van Schalkwyk, D.J.; Van der Watt, J.J.; Vincent, T.J.; Purchase, I.F. Hepatocellular carcinoma and dietary aflatoxin in Mozambique and Transkei. Br J Cancer 1985, 51, 713-726, doi:10.1038/bjc.1985.107.

2. van Wyk, P.; Van der Merwe, P.; Subrahmanyam, P.; Boughton, D. Aflatoxin contamination of groundnuts in Mozambique. International Arachis Newsletter 1999, 19, 25-27.

Author Response

Dear reviewer,

Thank you so much for your critical review of the manuscript submitted for publication in IJERPH and for your specific comment that I considered with care to address them the best way possible.

Please find a point-by-point response to each of your comments and the correspong alteration in the revised manuscripts (original munusript divided into two manuscripts in the revisied bersion)

Best ragrds

N. benkerroum

Response

Reviewer 1
The author wrote a respectable review article on aflatoxins, arguably the most complete to date.
It is more extensive than conventional reviews but it does not seem to be a problem because the the guidelines does not limit the number of pages.
Response:
Thanks for the compliments. However, another reviewer suggests splitting the manuscript into two parts each of which stands alone for clarity and coherence, and the present manuscript was revised accordingly.
Manuscript 1: Aflatoxins: A Comprehensive Overview Production, Structure, Health Issues and Incidence in Southeast Asian and Sub-Saharan African Countries
Manuscript 2: Chronic and Acute Toxicities of Aflatoxins: Mechanisms of Action
In terms of content fluency, it was quite well-written, but there are some structural issues, overlooked journal guidelines and tips to be followed with potential to improve the work:
Response:
Some language adjustments were made; the style will be amended during the editorial process to meet strictly the journal’s guidelines
The abstract is too extensive. According to the guidelines, it "should be a total of about 200 words maximum";
Response:
Abstracts revised as per to your suggestions. Thanks
There are plenty of blank spaces before and after the tables. Eliminate them, as they compromise the esthetic. It is not a problem to bring up as much text as possible and fill these gaps. There are exceptional cases when you use "landscape" pages, but in regular situations avoid blank spaces;
Response:
The present format aimed to help reviewers reading clearly the table and figures. Upon final acceptance, the technical editor will adjust the typesetting of the paper in more professional manner.
Some tables are too large. Some information could be sent as supplementary material. Most information is important for reading or studying but some is just relevant for consultation, confirmation or curiosity purposes. For instance, some aflatoxins are not as concerning as the well-known types B, G and M. You could send the remaining as supplementary material and refer in the caption or table footnote;
Response:
Done as per your recommendation.
Table 2: Reduced and the complete table was used as Tables 1S supplementary material
Tables 5 and 6: Summarized and the completes versions used as Table 2S and Table 3S supplementary materials
Keep figures and captions on the same page (e.g. figure 1). You can manipulate the spaces between the images or their sizes to achieve it;
Response:
This is also a style issue that will be solved during the journal’s typesetting of the final versions
Table 3: you should mention Mozambique because the country is historically relevant for aflatoxin research. It was there where Van Rensburg [1] conducted a pioneering study on the epidemiological association between aflatoxin intake and hepatocellular carcinoma, and later also van Wyk [2] presented an important article raising awareness on the need to reduce aflatoxins in groundnuts from Southern Africa in order to be able to export them to Europe. In Southern Africa, the latter study represent the turning point in aflatoxin research, from mostly a public health issue to an agricultural concern. The third turning point was certainly the 2004 outbreak in Kenya properly mentioned in your study. Considering the broadness of your review's scope, these studies seem important to mention to provide some context. Furthermore, there are more recent publications on aflatoxins in Mozambique.
Response:
Thank you for the references and the observation. Data on Mozambique were added as suggested in the summarized Table 5 of the revised version (Manuscript 1) and in the corresponding supplementary table S2.
Avoid as much as you can writing URL in the text as replacement to citations. For instance, you did that in table 2's caption. Write the full reference in the reference list and cite it in text just like any other source; Revise the page numbering;
Response:
URL references changes where appropriate to reference citations.
Figure 4's caption is too extensive. Consider putting some of its content in the main text when discussing about the figure. Fit the caption within one page.
Response:
This will be taken care of in the final version. Unless required, transferring the captions to the text will be cumbersome and will probably lose the reader with too many specific molecular biology details. Yet the information is necessary for the clarity and interested readers may see all the details in the captions. Again, fitting the figure with its caption within one page may be possible during the typesetting of the final version.
Again, many thanks for having read the manuscript and for the constructive comments.
REFERENCE
1. Van Rensburg, S.J.; Cook-Mozaffari, P.; Van Schalkwyk, D.J.; Van der Watt, J.J.; Vincent, T.J.; Purchase, I.F. Hepatocellular carcinoma and dietary aflatoxin in Mozambique and Transkei. Br J Cancer 1985, 51, 713-726, doi:10.1038/bjc.1985.107.
2. van Wyk, P.; Van der Merwe, P.; Subrahmanyam, P.; Boughton, D. Aflatoxin contamination of groundnuts in Mozambique. International Arachis Newsletter 1999, 19, 25-27.

Reviewer 2 Report

I think that it is a complete work and endorse for the publication.

Author Response

Dear reviewer,

Thank you very much for the time to critically review the manuscript submitted for publication in IJERPH and for your positive comments.

Due to the excessive lenght of the original submission, another reviewer suggested to divide it into two new manuscript, which was done in this new submission.

Kind regards

N. Benkerroum

Reviewer 3 Report

General comments

While the author is complimented in drafting such a comprehensive review on Aflatoxins, the ambition to present all different aspect of this subject, has resulted in an extremely long manuscript which loss coherence and focus. In its current form, the manuscript has no clear message other than the very generic statements that more research is needed. This is not very convincing given the 80.800 publications on Aflatoxins that are indexed in Goggle Scholar in December 2019!

In conclusion, this reviewer strongly suggests dividing this manuscript in at least 2 part, which should be published separately, using the following comments as a guidance for the authors.

Part 1 (Major revision)

Chapter 2. : Production, Structural Diversity, and Main Toxicological Properties of Aflatoxins.

2.1. The focus here lays on toxin producing fungal species and structural diversity. This part is valuable as a single subject

Lines 110-140 address the possibility of an introduction of atoxicogenic fungal strains (competitive exclusion principle). This is an important measure (mentioned also in the overall conclusions) to mitigate the aflatoxin risk. However, currently the importance of the subject is almost invisible in the text and this part should be moved (see suggestion below).

2.2. Table 1: This table is partly misleading and needs to be splitted to more clearly show the differences between fungal metabolites, bacterial metabolites (as for example aflatoxicol which is mainly produced by intestinal microbiota) and animal (biotransformation – derived) metabolites (such as AFM1, AFP1, AFQ1 etc). Moreover, the route of contamination should be more clearly described for individual metabolites.  A typical example is dairy milk, which might be contaminated with AFB1, when milk or dairy products are spoiled and invaded with fungal species, versus contamination with AFM1, originated from hepatic metabolism of dairy cows having consumed aflatoxin contaminated feed.   Although many of theses details are correctly described later in the manuscript, this table needs revision to avoid misinterpretation.

Aflatoxin production

The description of aflatoxin production (text part) is a valuable summary. However, table 5 should be revisited and amended, as in its current form, is it an unhappy mixture of data sorted according climatic regions, provinces and vegetation zones. Please try to harmonize, using the text that is now given in the legend of the table. Moreover, all these data are only examples of contamination levels. Therefore, it is recommended to condense the data (in a much shorter table) as individual data from individual provinces do not contribute to the overall understanding to contamination levels. Considering that human consumption of the different commodities remains unknown, the data are not suitable for an exposure assessment.

Moreover, this part should be preceded with a short description of the methods of detection of aflatoxins applied in the different studies.

Feed contamination (including table 7): It is recommended to delete this table here as the given data are very incomplete (see recent reviews produced by the feed industry) and add no relevant information to this part of the manuscript. The authors are strongly advised to keep in the entire manuscript the focus on human exposure and diseases associated with aflatoxin exposure.

Aflatoxin Toxicity: The description of outbreaks of aflatoxicosis in humans in lengthy and partly anecdotal. The value of this information is rather limited as the actual exposure rate (dose aflatoxin ingested) remains unclear. Subsequently there is no link between these empirical data and the mechanism of toxicity described in the next part (which therefore should be separated).

To complete this first part, it is recommended to integrate Chapter 9 from the current manuscript (human data) and the findings reported in Chapter 8 (8.1. and 8.2.) and partly in Chapter 7 (epidemiological data) into this part (as they contain valuable information but are misplaced in the second part).

Overall recommendation:

Part 1 (the first manuscript) should end here and needs to be completed with a conclusion part. Here mitigation strategies to reduce human exposure would be a valuable addition, including the information given now in lines 110-140, which could be presented here in a prominent way. Finallt, a brief review of the current risk assessment approaches (exposure assessment, health-based guidance levels as presented by the international organizations such as WHO or EFSA) would perfectly complete this first part.

Part 2:

Mechanisms of Toxicity of aflatoxins (minor revision)

This part should be critically revisited also. While it is beyond this first review to address all details, the authors are invited to consider that readability could be improved by

A clear differentiation between in vitro data and in vivo findings at all places (perhaps even by introducing more subchapters (such as genetic polymorphism, effects on the cell cycle etc – comparable to the section about the Immune system) A clear mentioning of the doses/concentration used in the individual studies (replacing the too often used generic terms such as high or low doses).

Finally, part 2 should be completed with a (new) critical discussion/conclusion section

Author Response

Dear reviewer,

Thank you so much for your time to review critically the manuscript submitted for publication to IJERPH and for providing valuable comments and juggestions to improve the quality and impact of the review.

Your comments were considred with care and addressed the best way possible. Please find below a summary of a point-by-point response to each of your comments, while the corresponding alterations are highlighted in the new manuscripts in blue font color by the track change facility of microsoft word.

Kind regards

N. Benkerroum

Point-by-point response

General comments
While the author is complimented in drafting such a comprehensive review on Aflatoxins, the ambition to present all different aspect of this subject, has resulted in an extremely long manuscript which loss coherence and focus. In its current form, the manuscript has no clear message other than the very generic statements that more research is needed. This is not very convincing given the 80.800 publications on Aflatoxins that are indexed in Goggle Scholar in December 2019!
Response:
Thank you for the observation. However, I acknowledged in a previous paper in the same issue the tremendous work that has been done on aflatoxins. Yet, there are still many obscure aspects that hinder practical actions that I tried to mention in the text whenever justified as part of the message the review should convey. Despite the impressive annual numbers of publications on aflatoxins, the mechanism of action is still poorly understood; e.g., the synergy between hepatitis virus infection although beyond doubt is not explicitly explained, the present knowledge of immunotoxicity does not allow curative or preventive actions, neurotoxicity, etc. I thought that part of the review objective is to highlight such gaps.
Nonetheless, I hope that the present version of two separate parts provides clearer messages
In conclusion, this reviewer strongly suggests dividing this manuscript in at least 2 part, which should be published separately, using the following comments as a guidance for the authors.
Response:
Manuscript divided into two parts:
Manuscript 1: Aflatoxins: A Comprehensive Overview Production, Structure, Health Issues and Incidence in Southeast Asian and Sub-Saharan African Countries
Manuscript 2: Chronic and Acute Toxicities of Aflatoxins: Mechanisms of Action
Part 1 (Major revision)
Chapter 2. : Production, Structural Diversity, and Main Toxicological Properties of Aflatoxins.
2.1. The focus here lays on toxin producing fungal species and structural diversity. This part is valuable as a single subject
Lines 110-140 address the possibility of an introduction of atoxicogenic fungal strains (competitive exclusion principle). This is an important measure (mentioned also in the overall conclusions) to mitigate the aflatoxin risk. However, currently the importance of the subject is almost invisible in the text and this part should be moved (see suggestion below).
Response:
Done. Manuscript 1: Lines 879-964
2.2. Table 1: This table is partly misleading and needs to be splitted to more clearly show the differences between fungal metabolites, bacterial metabolites (as for example aflatoxicol which is mainly produced by intestinal microbiota) and animal (biotransformation – derived) metabolites (such as AFM1, AFP1, AFQ1 etc). Moreover, the route of contamination should be more clearly described for individual metabolites. A typical example is dairy milk, which might be contaminated with AFB1, when milk or dairy products are spoiled and invaded with fungal species, versus contamination with AFM1, originated from hepatic metabolism of dairy cows having consumed aflatoxin contaminated feed. Although many of theses details are correctly described later in the manuscript, this table needs revision to avoid misinterpretation.
Response:
While I strongly agree with your observations; However, I split aflatoxins according to their structural grouping (Difurocoumarocyclopentenone and Difurocoumarolactone) to be coherent with the text, while paying attention to the completeness of information on each aflatoxin, so that interested readers by a given aflatoxin can find succinct by enough information in Table 1.
Aflatoxin production
The description of aflatoxin production (text part) is a valuable summary. However, table 5 should be revisited and amended, as in its current form, is it an unhappy mixture of data sorted according climatic regions, provinces and vegetation zones. Please try to harmonize, using the text that is now given in the legend of the table. Moreover, all these data are only examples of contamination levels. Therefore, it is recommended to condense the data (in a much shorter table) as individual data from individual provinces do not contribute to the overall understanding to contamination levels. Considering that human consumption of the different commodities remains unknown, the data are not suitable for an exposure assessment.
Response
As per your recommendation and that of another reviewer, summarized Tables 4, 5 and 6 were produced, while the original tables with the complete data were used as supplementary materials Tables S1, S2 and S3 in the present manuscript 1, in case readers are interested by details
Moreover, this part should be preceded with a short description of the methods of detection of aflatoxins applied in the different studies.
Response:
A short description of the main methods used to produce the data was added. Yet, the description was kept minimal since it was detailed in the previously published manuscript in the same issue to avoid redundancy. Lines 273-278.
Feed contamination (including table 7): It is recommended to delete this table here as the given data are very incomplete (see recent reviews produced by the feed industry) and add no relevant information to this part of the manuscript. The authors are strongly advised to keep in the entire manuscript the focus on human exposure and diseases associated with aflatoxin exposure.
Response:
Table 7 deleted, although the text was left to give a brief idea on the impact of feed contamination with aflatoxins on human health.
Aflatoxin Toxicity: The description of outbreaks of aflatoxicosis in humans in lengthy and partly anecdotal. The value of this information is rather limited as the actual exposure rate (dose aflatoxin ingested) remains unclear. Subsequently there is no link between these empirical data and the mechanism of toxicity described in the next part (which therefore should be separated).
To complete this first part, it is recommended to integrate Chapter 9 from the current manuscript (human data) and the findings reported in Chapter 8 (8.1. and 8.2.) and partly in Chapter 7 (epidemiological data) into this part (as they contain valuable information but are misplaced in the second part).
Response:
Done. Manuscript 1. Lines 475-524
Overall recommendation:
Part 1 (the first manuscript) should end here and needs to be completed with a conclusion part. Here mitigation strategies to reduce human exposure would be a valuable addition, including the information given now in lines 110-140, which could be presented here in a prominent way. Finallt, a brief review of the current risk assessment approaches (exposure assessment, health-based guidance levels as presented by the international organizations such as WHO or EFSA) would perfectly complete this first part.
Response:
Done. Manuscript 1:
Lines 720-877 (Risk assessment)
Lines 879-964: Mitigation strategies
Part 2:
Mechanisms of Toxicity of aflatoxins (minor revision)
This part should be critically revisited also. While it is beyond this first review to address all details, the authors are invited to consider that readability could be improved by
A clear differentiation between in vitro data and in vivo findings at all places (perhaps even by introducing more subchapters (such as genetic polymorphism, effects on the cell cycle etc – comparable to the section about the Immune system) A clear mentioning of the doses/concentration used in the individual studies (replacing the too often used generic terms such as high or low doses).
Response
Two section headings relative to polymorphism and cell cycle have been added.
Line 211: Genetic Polymorphism and Increased Mutagenicity of Aflatoxins
Line 260: Cell Cycle Progress as Affected by Aflatoxin-induced p53-gene Mutation
Finally, part 2 should be completed with a (new) critical discussion/conclusion section
Response:
Done.

Round 2

Reviewer 3 Report

The author is complemented with the revised manuscript.